# Diminished Short- and Long-Term Antibody Response after SARS-CoV-2 Vaccination in Hemodialysis Patients

**DOI:** 10.3390/vaccines10040605

**Published:** 2022-04-13

**Authors:** Louise Füessl, Tobias Lau, Isaac Lean, Sandra Hasmann, Bernhard Riedl, Florian M. Arend, Johanna Sorodoc-Otto, Daniela Soreth-Rieke, Marcell Toepfer, Simon Rau, Haxhrije Salihi-Halimi, Michael Paal, Wilke Beuthien, Norbert Thaller, Yana Suttmann, Gero von Gersdorff, Ron Regenauer, Anke von Bergwelt-Baildon, Daniel Teupser, Mathias Bruegel, Michael Fischereder, Ulf Schönermarck

**Affiliations:** 1Department of Medicine IV, University Hospital, LMU Munich, 81377 Munich, Germany; louise.fueessl@med.uni-muenchen.de (L.F.); isaac.lean@med.uni-muenchen.de (I.L.); sandra.hasmann@med.uni-muenchen.de (S.H.); ron.regenauer@med.uni-muenchen.de (R.R.); anke.bergwelt@med.uni-muenchen.de (A.v.B.-B.); michael.fischereder@med.uni-muenchen.de (M.F.); 2Dialysezentrum Bad Tölz und Wolfratshausen, 83646 Bad Tölz, Germany; telau@gmx.de (T.L.); simonrau@hotmail.com (S.R.); 3KfH-Nierenzentrum Bayreuth, 95445 Bayreuth, Germany; bernhard.riedl@kfh.de (B.R.); haxhrije.salihi-halimi@kfh.de (H.S.-H.); 4Institute of Laboratory Medicine, University Hospital, LMU Munich, 81377 Munich, Germany; florian.arend@med.uni-muenchen.de (F.M.A.); michael.paal@med.uni-muenchen.de (M.P.); daniel.teupser@med.uni-muenchen.de (D.T.); mathias.bruegel@med.uni-muenchen.de (M.B.); 5KfH-Nierenzentrum Germering, 82110 Germering, Germany; johanna.sorodoc-otto@kfh.de (J.S.-O.); wilke.beuthien@kfh.de (W.B.); yana.suttmann@kfh.de (Y.S.); 6KfH-Nierenzentrum Miesbach, 83714 Miesbach, Germany; daniela.soreth-rieke@kfh.de (D.S.-R.); norbert.thaller@kfh.de (N.T.); 7Dialysezentrum Garmisch-Partenkirchen-Murnau-Weilheim, 82418 Murnau, Germany; mtoepfer@gmx.eu; 8QiN-Group, Department II of Internal Medicine, Faculty of Medicine and University Hospital Cologne, University of Cologne, 50935 Cologne, Germany; gero.freiherr-von-gersdorff@uk-koeln.de

**Keywords:** SARS-CoV-2, vaccination, antibody response, hemodialysis

## Abstract

Short-term studies have shown an attenuated immune response in hemodialysis patients after COVID-19-vaccination. The present study examines how antibody response is maintained after vaccination against SARS-CoV-2 in a large population of hemodialysis patients from six outpatient dialysis centers. We retrospectively assessed serum antibody levels against SARS-CoV-2 spike protein and nucleocapsid protein (electrochemiluminescence immunoassays, Roche Diagnostics) after COVID-19-vaccination in 298 hemodialysis and 103 non-dialysis patients (controls), comparing early and late antibody response. Compared to a non-dialysis cohort hemodialysis patients showed a favorable but profoundly lower early antibody response, which decreased substantially during follow-up measurement (median 6 months after vaccination). Significantly more hemodialysis patients had anti-SARS-CoV-2-S antibody titers below 100 U/mL (*p* < 0.001), which increased during follow-up from 23% to 45% but remained low in the control group (3% vs. 7%). In multivariate analysis, previous COVID-19 infections (*p* < 0.001) and female gender (*p* < 0.05) were significantly associated with higher early as well as late antibody vaccine response in hemodialysis patients, while there was a significant inverse correlation between patient age and systemic immunosuppression (*p* < 0.001). The early and late antibody responses were significantly higher in patients receiving vaccination after a SARS-CoV-2 infection compared to uninfected patients in both groups (*p* < 0.05). We also note that a higher titer after complete immunization positively affected late antibody response. The observation, that hemodialysis patients showed a significantly stronger decline of SARS-CoV-2 vaccination antibody titers within 6 months, compared to controls, supports the need for booster vaccinations to foster a stronger and more persistent antibody response.

## 1. Introduction

In the ongoing COVID-19 pandemic, dialysis patients represent a particularly vulnerable population with a higher rate of morbidity and mortality than the general population. Thus, dialysis patients were prioritized in global vaccination strategies.

While several groups have shown a favorable overall short-term response rate after two vaccinations, the magnitude of the humoral response is significantly lower than in health care workers or non-dialysis patients [1,2]. Factors associated with a lower or absent immune response are older age, immunosuppressive therapy, and the use of the BNT162b2 compared to the mRNA-1273 vaccine. In contrast, hemodialysis patients with a prior infection mount a substantially higher antibody response after vaccination [3,4,5]. Despite this diminished early antibody response in dialysis patients after vaccination against SARS-CoV-2, this strategy has proven to prevent severe COVID-19 infection [6].

However, as is described after natural infection [7,8], concerns remain about the faster waning of antibody levels after vaccination in this group. Extended follow-up data on the humoral antibody response after COVID-19 in dialysis patients is scarce. Only a few studies have reported data on the longevity of the humoral response to vaccination, demonstrating a rapid decline of antibody response 3 to 6 months after vaccination [9,10,11,12,13,14]. Furthermore, patients with lower antibody titers seem to be at higher risk for COVID-19 breakthrough infections [9,15].

Therefore, measuring the immune response after vaccination against SARS-CoV-2 in dialysis patients may aid in the identification of dialysis patients at risk for infection and thus guide clinical management. In this retrospective study, we examined the early and late antibody immune response after vaccination against SARS-CoV-2 in a large cohort of hemodialysis patients with and without previous COVID-19 infection in comparison to patients not requiring dialysis. To evaluate potential confounders, demographics, comorbidities, and use of immunosuppressive medication were assessed.

## 2. Materials and Methods

### 2.1. Study Setting

The present study took place in six different outpatient dialysis centers. We retrospectively analyzed individuals aged ≥18 years who received full COVID-19 vaccination in accordance with the national recommendations at that time (two vaccinations with either mRNA or vector-based vaccine; or one vaccination with mRNA vaccine after prior documented COVID-19 infection) and who had SARS-CoV-2 antibody response measured. Patients with a third vaccination during the study period or incomplete data were excluded. We did not exclude patients with prior COVID-19 infection or immunosuppressive therapy from the analysis. The control group consisted of patients without dialysis treated at the same outpatient centers (patients with chronic kidney disease, patients on LDL apheresis, kidney transplant patients, and health care workers during their regular medical visits).

### 2.2. Ethics

The study was approved by the local institutional review board of the LMU Munich (No. 21-0960). Several patients were included in a previous retrospective study reporting early humoral response to SARS-CoV-2-vaccination [5].

### 2.3. Laboratory Testing

SARS-CoV-2 antibody testing was performed with electrochemiluminescence immunoassays designed to detect antibodies (including IgG) against the SARS-CoV-2 S (spike) protein (Elecsys Anti-SARS-CoV-2 S, Roche Diagnostics, Mannheim, Germany) and antibodies against the SARS-CoV-2 N (nucleocapsid) protein (Elecsys Anti-SARS-CoV-2, Roche Diagnostics, Mannheim, Germany). Seroconversion in SARS-CoV-2 infection yields antibodies targeting both the spike and nucleocapsid proteins, while SARS-CoV-2 vaccination (without previous infection) only leads to the presence of antibodies against the spike protein. Testing was performed in the Institute of Laboratory Medicine of the University Hospital Munich. According to the manufacturer’s specifications, Anti-SARS-CoV-2 S antibody titers ≥ 0.8 U/mL are considered reactive (sensitivity 98.8%, specificity 99.9%). A titer of <100 U/mL was defined as low titer.

### 2.4. Data Evaluation

Due to the high rate of asymptomatic SARS-CoV-2 histories, patients with a previously positive PCR result before vaccination, as well as positive antibody reactions to the SARS-CoV-2 N protein, were considered previously infected. The evolution of anti-SARS-CoV-2 S antibody titers over time was studied. For this purpose, two time periods were defined, based on the time between vaccination and measurement of anti-SARS-CoV-2 S antibody titers: T1 = early response with antibody measurement <100 days after complete vaccination and T2 = follow-up response > 100 days after complete vaccination. Blood investigations for measurement of anti-SARS-CoV-2 S antibody titers were performed between March and September 2021.

### 2.5. Statistical Analysis

Differences in anti-SARS-CoV-2 S antibody titers were analyzed using a Mann–Whitney–U test. Patients’ characteristics were compared between groups using a Mann–Whitney–U test for numerical data and a Fisher’s exact test (two-tailed) for categorical data. *p* values < 0.05 were considered significant. Multivariate linear regression analysis and multivariate analyses for variance (MANOVA) with log10 transformed anti-SARS-CoV-2 S antibody titers as outcomes were performed with age, sex, COVID-19 infection (previous and after vaccination), use of systemic immunosuppressive therapy, and time between vaccination and antibody measurement as independent variables. Statistical analysis was performed with EasyMedStat^®^ (version 3.15.1; www.easymedstat.com (accessed on 13 February) and MATLAB^®^ (version R2021b, 2021, Natick, MA, USA).

## 3. Results

### 3.1. Demographic and Clinical Data

The study population was divided into a case group consisting of 298 hemodialysis patients and a control group of 103 patients without dialysis (consisting of patients with chronic kidney disease, patients on LDL apheresis, kidney transplant patients, and health care workers during their regular medical visits). Approximately 11% of the hemodialysis patients and 9% of the control group had confirmed evidence of SARS-CoV-2 infection before vaccination.

From a total of 401 vaccinated patients, 381 were immunized with the BNT162b2 vaccine (Pfizer-BioNTech, Mainz, Germany), 5 with the mRNA-1273 vaccine (Moderna, Cambridge, MA, USA), 10 with the vector-based ChaAdOx1 nCoV-19 vaccine (Oxford-Astra Zeneca, Cambridge, UK), and 5 patients received a mixed vaccine protocol, without significant difference between both cohorts.

Vaccinated hemodialysis patients had been on dialysis treatment with a median of 43.5 months, the median age was 73 years, and 64% of the hemodialysis patients were male. Detailed patients’ characteristics can be found in Table 1.

COVID 19 infection occurred in 34 hemodialysis patients and 9 controls before vaccination. 13 out of 34 hemodialysis patients with a COVID infection before vaccination were admitted to the hospital, 4 of them to the intensive care unit. Four patients contracted a COVID infection after complete vaccination. All infections occurred before the emergence of the Omicron variant.

### 3.2. SARS-CoV-2 Antibody Response

#### 3.2.1. Early Humoral Antibody Response (T1)

The majority of hemodialysis patients and controls developed a detectable early humoral antibody response (T1; anti-SARS-CoV-2 S antibody titer ≥ 0.8 U/mL) measured less than 100 days after completion of vaccination (97.1% vs. 97.8%) (Table 1). Uninfected control patients showed significantly higher anti-SARS-CoV-2 S antibody titers compared to uninfected hemodialysis patients (median (Q1–Q3) 1737.5 (838–2406) vs. 265 (100; 706) U/mL, *p* < 0.001). Overall, 23% of the hemodialysis patients presented with a low antibody response (<100 U/mL) as opposed to 3% of the control group (*p* < 0.001). The best antibody response was observed in patients receiving vaccination after a previous SARS-CoV-2 infection. The results proved to be significantly higher compared to uninfected patients in both groups (*p* < 0.05). In contrast, the median anti-SARS-CoV-2 S antibody titer in hemodialysis patients with systemic immunosuppressive therapy was even lower (10.8 U/mL).

#### 3.2.2. Late Humoral Antibody Response (T2)

After a median of 6 months (T2; range 3.3–7 months), anti-SARS-CoV-2 S antibody titers decreased in most patients, remained however detectable (anti-SARS-CoV-2 S antibody titer ≥ 0.8 U/mL) in the majority of hemodialysis patients and controls (96.8% vs. 100%) (Table 1 and Figure 1). Anti-SARS-CoV-2 S antibody titers decreased in both hemodialysis patients and controls. However, they were significantly lower in uninfected hemodialysis patients compared to uninfected controls (median (Q1–Q3) 101.5 (29–240) vs. 469.5 (307–865) U/mL, *p* < 0.001). The late antibody response was significantly higher in patients receiving vaccination after SARS-CoV-2 infection compared to uninfected patients in both groups (*p* < 0.05).

Generally, the decline of the anti-SARS-CoV-2 S antibody titers tended to be faster in hemodialysis patients compared to non-dialysis controls (Figure 2). The percentage of hemodialysis patients with a titer below 100 U/mL increased up to 45% but remained low in the control group (7%) (*p* < 0.001).

Among uninfected patients with antibody measurements at T1 and T2 (*n* = 309), we observed a comparable relative decrease in antibody levels (median, −60% in hemodialysis patients vs. −66% in control patients). The relative decrease was similar in hemodialysis patients with COVID-19 infection before vaccination (median, −69%). During the observation period, four hemodialysis patients contracted a COVID-19 infection (two of them within 2 months after complete vaccination). Among these patients, anti-SARS-CoV-2 S antibody titers substantially increased (median T1: 134.5 U/mL, T2: 3704.5 U/mL). Although 12 hemodialysis patients died during the observation period, there was no COVID-19 related death recorded.

### 3.3. Multivariate Analysis

Among hemodialysis patients, previous COVID-19 infection and female gender were significantly associated with higher values of anti-SARS-CoV-2 S antibody levels at both short-term (T1) and long-term (T2) follow-up (Table 2). There was also a significant inverse correlation between the humoral antibody response, systemic immunosuppressive therapy, and patients’ age. The negative association with immunosuppressive therapy, age, and male gender could also be shown when looking at hemodialysis patients without prior COVID-19 infection (data not shown).

When incorporating early anti-SARS-CoV-2 S antibody titers (T1) into multivariate regression analysis of late antibody response (T2), only early anti-SARS-CoV-2 S antibody titer and previous COVID-19 infection remained significantly associated with late anti-SARS-CoV-2 S antibody titers.

Patients in the hemodialysis group were significantly older than controls. In a regression analysis, age could be excluded as a potential confounder.

## 4. Discussion

Patients undergoing maintenance hemodialysis are generally immunocompromised, resulting in an impaired response to vaccinations, such as hepatitis B. Additional risk factors for a low antibody response in these patients are immunosuppressive therapy and previous chemotherapy.

Our real-world data in a large cohort of hemodialysis patients demonstrate that COVID-19 vaccines, unlike hepatitis B and influenza vaccination, elicit a substantial antibody response in the dialysis population. However, the humoral antibody response after vaccination is significantly lower than in non-dialysis patients or health care workers. During a median follow-up of 6 months after vaccination, we observed a further substantial decrease in SARS-CoV-2 S antibody titers leading to low antibody levels (<100 U/mL) in nearly half of the hemodialysis population.

Numerous studies have shown similar results early after vaccination with a high seroconversion rate of hemodialysis patients early after vaccination, but achieving significantly lower antibody levels than non-dialysis cohorts [1,2,13]. So far, only a few studies have reported extended follow-up data. While the number of seronegative patients remained low during long-term follow-up in our cohort (3%), other studies reported a decrease in seroconversion from 95–98% to 66–81% [11,13,14,15,16]. Accordingly, antibody titers significantly decreased in all studies. The median antibody declines of 55% [17], 56% [15], and 84% [18] 3 to 6 months after vaccination correspond well to our data (60%).

The gradual waning of antibody levels has also been reported in the general population. Even with a rate of decline that is similar to that of the general population, hemodialysis patients with low early antibody levels will lose their protective response vis-à-vis the control group faster. In our study, the number of hemodialysis patients with low antibody titers (<100 U/mL) increased from 23% to 45% at long-term follow-up. Although a minimally required threshold for clinical protection is currently unknown, a correlation between antibody response and protection against infection has been observed in the general population [19,20]. Boudhabhay et al. showed in a small study of hemodialysis patients that antibody titers of hemodialysis patients measured before exposure correlated with protection from SARS-CoV-2 infection [21]. However, the protective threshold might vary among virus variants [20].

In our multivariate analysis, age, male gender, and immunosuppressive therapy were associated with a lower early as well as late SARS-CoV-2 antibody response. This finding is supported by other recently published studies [10,11,13,22]. In contrast, hemodialysis patients with COVID-19 infection before or after vaccination showed a more pronounced long-term vaccine response leading to antibody titers equal to non-dialysis vaccinated individuals. This booster effect has been described in health care workers with preceding infection and in hemodialysis patients [10,18]. Recently, in a study of health care workers, Wratil et al. showed that three consecutive spike antigen exposures, either as triple immunization or infection-plus-vaccination resulted in an improved antibody response with superior neutralization capacity against variants of concern [23].

In previous studies, vaccination with the mRNA-1273 vaccine resulted in higher short- and long-term antibody responses compared to the BNT162b2 vaccine [13,24,25,26], which may be related to its higher mRNA dose. Overall, our data support the observation of a more durable long-term antibody response, when the initial antibody titers are higher [13].

In general, an altered and weaker immune response has been shown for other vaccines like influenza or hepatitis B [27,28]. Regarding the latter, this led to strategies of identifying individuals at risk by regular measurement of antibody status and of using increased doses or booster vaccinations [29]. A correlation between responsiveness to hepatitis B vaccination and seroconversion after SARS-CoV-2 vaccination could not be demonstrated so far [14,16]. It remains unclear if the spike protein is more immunogenic than the HBs antigen or if mRNA vaccines cause increased immunogenicity.

The strengths of our study are the multi-center design and the large number of hemodialysis patients within a real-world setting. However, it was not possible to evaluate the impact of different vaccines as the majority of patients received the BNT162b2 vaccine. A further limitation of the study is that we could not assess cellular immunity (especially memory T cells), which strongly contributes to the longevity of immunity against SARS-CoV-2 and can also be found in patients under immunosuppressive therapy without detectable antibody response. Recent studies suggest that in addition to the humoral response, the T cell response is also compromised in hemodialysis patients [26,30]. However, the measurement of anti-SARS-CoV-2 antibodies is implemented in routine clinical care and provides an easy and affordable way to monitor the humoral immune response and identify vulnerable patients at high risk of poor outcomes.

During the study period, the incidence of COVID-19 infections in Germany was low and the Omicron variant was not yet prevalent. Protection against the new, emerging SARS-CoV-2 variants may require even higher antibody levels. As a result of the timing of this study, we could not evaluate the true impact of antibody levels on the infection risk, especially with new variants of concern.

Vaccination is associated with a lower risk of COVID-19 infection in hemodialysis patients and with a reduced risk of hospitalization or death among those diagnosed with COVID-19 [6]. Given the waning immune response, with half of the hemodialysis patients developing low or undetectable antibody titers 6 months after vaccination, there is a need for a booster vaccination in this highly vulnerable population, in line with global recommendations. This would be consistent with our finding that dialysis patients with a prior infection had a much higher antibody response than their vaccine naïve companions. However, the early- and long-term antibody response is highly variable both in quantity and duration. Given the high-risk dialysis patients face in case of a COVID-19 infection, we argue in favor of regular assessment of quantitative antibody titers over time to identify patients at risk and as such to determine the optimal time for booster vaccinations. Similar testing strategies might also be useful for other vaccinations. However, further investigation is needed to define protective thresholds to individualize vaccination strategies.

## 5. Conclusions

In summary, we report a declining antibody response 6 months after vaccination against SARS-CoV-2 in hemodialysis patients, except for the immune response of hemodialysis patients with prior COVID-19 infection. This argues in favor of a booster vaccination and suggests that measurement of the antibody response may be of clinical utility. This is especially true in patients with low or no response and therefore a high risk of an infection who might in turn benefit from regular antibody testing over time or intensified vaccine schedules. Further studies should evaluate protective thresholds and the antibody response after booster vaccinations.

## Figures and Tables

**Figure 1 vaccines-10-00605-f001:**
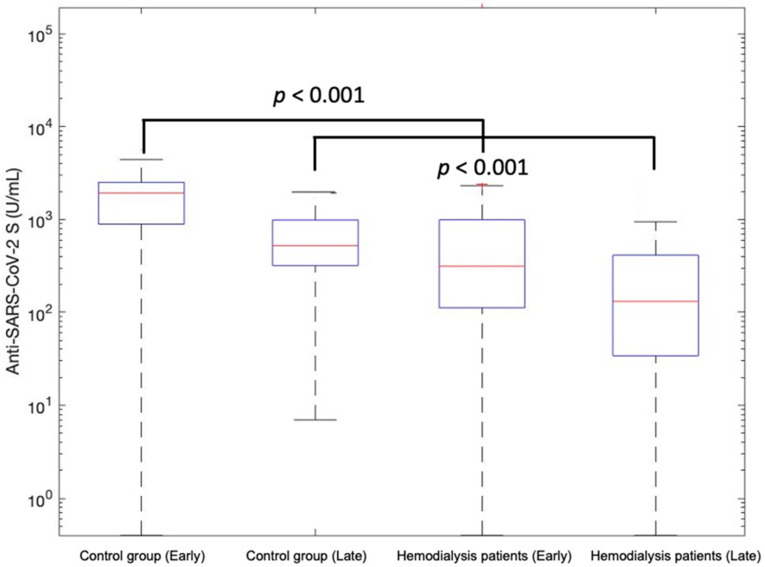
Early and late Anti-SARS-CoV-2 S antibody response in individuals after COVID-19 vaccination. Anti-SARS-CoV-2 S antibody titers are shown in uninfected hemodialysis patients and controls. The box shows the interquartile range, the horizontal line inside the box represents the median values, and whiskers represent the minimum and maximum range of points within 1.5 times the interquartile range in the box.

**Figure 2 vaccines-10-00605-f002:**
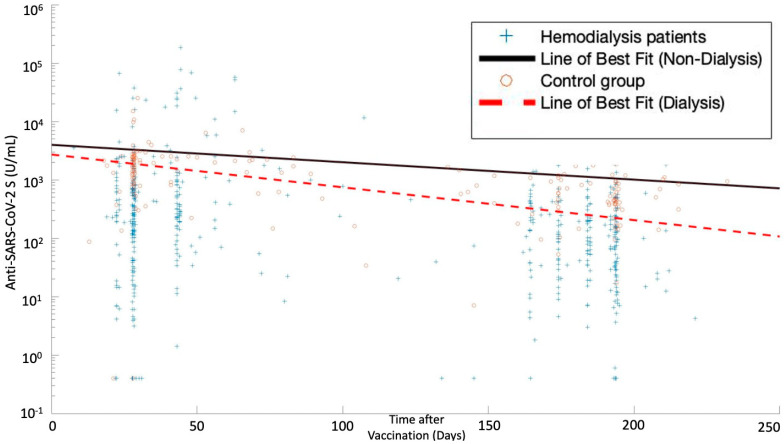
Anti-SARS-CoV-2 S antibody titers versus time after completion of vaccination in hemodialysis patients and controls. The symbols represent individual anti-SARS-CoV-2 S antibody measurements of hemodialysis patients (o) or controls (+). Dynamic profiling utilizing a scatter plot of data points with linear regression analysis of Anti-SARS-CoV-2 S antibody titers in patients undergoing hemodialysis and those not undergoing hemodialysis overtime was performed. We observed that the line of best fit for patients not undergoing hemodialysis and those undergoing hemodialysis is represented by the equation y = −10.38x + 3965 and y = −13.01x + 2702. Drawing from the abovementioned equations, we observe that (1) the titers generally decrease over the given period of time and (2) the titers in patients undergoing hemodialysis decrease more rapidly in comparison to those not undergoing hemodialysis (mHemodialysis = −13.01 vs. mControl = −10.38).

**Table 1 vaccines-10-00605-t001:** Patient characteristics.

Parameter	Patients with	Patients without	*p*-Value †
	Hemodialysis (*n* = 298)	Dialysis (*n* = 103)	
Age—year, median (Q1; Q3)	73 (58; 80)	54 (42; 60)	<0.001
Male gender, *n* (%)	191 (64.1)	23 (22.3)	<0.001
SARS-CoV-2 infection prior to vaccination, *n* (%)	34 (11.4)	9 (8.7)	0.58
SARS-CoV-2 infection after vaccination, *n* (%)	4 (1.3)	0	0.57
Anti-SARS-CoV-2 S early response			
≥0.8 U/mL, *n* (%)	272/280 (97.1)	91/93 (97.8)	1.00
<100 U/mL, *n* (%)	65/280 (23.2)	3/93 (3.2)	<0.001
Anti-SARS-CoV-2 S follow-up response			
≥0.8 U/mL, *n* (%)	241/249 (96.8)	87/87 (100)	0.118
<100 U/mL, *n* (%)	111/249 (44.6)	6/87 (6.9)	<0.001
Anti-SARS-CoV-2 S in uninfected patients			
Early response, median (Q1–Q3) (U/mL)	265 (100–706)	1737.5 (838–2406)	<0.001
Late response, median (Q1–Q3) (U/mL)	101.5 (29–240)	469.5 (307–865)	<0.001
Anti-SARS-CoV-2 S response in patients with COVID infection before vaccination			
Early response, median (Q1–Q3) (U/mL)	18,300 (5836–35,850)	6972 (2526–12,825)	0.079
Late response, median (Q1–Q3) (U/mL)	6886 (3361–4591)	1769 (1187–3187)	0.012
Anti-SARS-CoV-2 S response in patients with COVID infection after vaccination			
Early response, median (Q1–Q3) (U/mL)	134 (4.6–264)		n.a.
Late response, median (Q1–Q3) (U/mL)	3704.5 (2115–5294)		n.a.
History of cancer, *n* (%)	41 (13.8)	n.d.	n.a.
Diabetes, *n* (%)	91 (30.5)	n.d.	n.a.
Systemic immunosuppression, *n* (%)	15 (5.0)	13 (12.6)	0.013
Cumulative time on hemodialysis—mo, median (Q1–Q3)	43.5 (18.8–83.0)	n.a.	n.a.
BMI (kg/m^2^)—mean (SD)	26.9 (±5.7)	n.d.	n.a.

† Significance given by *p*-values was computed using the Mann–Whitney–U test for numeric data and Fisher’s exact test for categorical data. BMI, body mass index; mo, months; Q1, first quartile; Q3, third quartile; n.a., not applicable; n.d., not determined; SD, standard deviation.

**Table 2 vaccines-10-00605-t002:** Multivariate linear regression analyses of factors influencing the Anti-SARS-CoV-2 S antibody response to COVID-19 vaccination in hemodialysis patients.

Parameters	Early Humoral Antibody Response (*n* = 280)			Late Humoral Antibody Response (*n* = 249)		
	β Coefficient	95% CI	*p*-Value	β Coefficient	95% CI	*p*-Value
Age—year (risk for each 1-year increase)	−0.015	[−0.021; −0.009]	<0.001	−0.014	[−0.019; −0.0085]	<0.001
Female gender	0.242	[0.0571; 0.428]	0.0106	0.3	[0.119; 0.481]	0.0013
Previous SARS-CoV-2 infection	1.65	[1.28; 2.02]	<0.001	1.72	[1.38; 2.06]	<0.001
Systemic immunosuppression	−1.57	[−2.4; −0.73]	<0.001	−1.46	[−2,2, −0.727]	<0.001
Time between vaccination and measurement—d	0.0024	[−0.005; 0.0099]	0.525	−0.0002	[−0.0056; 0.0052]	0.942

CI, confidence interval; d, days.

## Data Availability

The data that support the findings of this study will be shared on reasonable request to the corresponding author.

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
