# Peer review of "Diminished Short- and Long-Term Antibody Response after SARS-CoV-2 Vaccination in Hemodialysis Patients"

_vaccines, 2022, doi:10.3390/vaccines10040605_

Round 1

Reviewer 1 Report

The manuscript addresses the long-term immune response against COVID-19 in a cohort of patients undergoing hemodialysis.

The topic is not new; however, the merit of the work consists precisely in evaluating a cohort with a more significant number of patients. Furthermore, the manuscript is well organized with a proper application of statistical tools to assess the immune response over approximately 6 months in these patients and a robust discussion of the immune response in patients submitted to hemodialysis.

The manuscript can be considered for publication in the present form. However, before we feel it is necessary to (1) inform that the immune response, whether assessed early or late, is IgG and (2) include a reference in line 261

Author Response

Reviewer 1:

We thank the reviewer for the valuable comments and suggestions.

The manuscript addresses the long-term immune response against COVID-19 in a cohort of patients undergoing hemodialysis.

The topic is not new; however, the merit of the work consists precisely in evaluating a cohort with a more significant number of patients. Furthermore, the manuscript is well organized with a proper application of statistical tools to assess the immune response over approximately 6 months in these patients and a robust discussion of the immune response in patients submitted to hemodialysis.

The manuscript can be considered for publication in the present form. However, before we feel it is necessary to (1) inform that the immune response, whether assessed early or late, is IgG and (2) include a reference in line 261

  • The used assay (Elecsys Anti-SARS-CoV-2 S) detects antibodies against the SARS-CoV-2 Spike (S) protein receptor-binding-domaine (RBD) exhibiting a good correlation with a surrogate virus neutralisation assay. However, the assay detects all immunoglobulin classes – including, but not exclusively, IgG. Referring to „IgG reponse“ only would therefore not be precise. Moreover, Pérez-Alós observed that the humoral immune response after a prior COVID infection not only consists of IgG but also of  IgA , the latter being the predominant immunoglobulin on mucous surface (Pérez-Alós et al. Modeling of waning immunity after SARS-CoV-2 vaccination and influencing factors. Nat Commun. 2022 Mar 28;13(1):1614. doi: 10.1038/s41467-022-29225-4).
  • We included a reference in line 285 (former line 261): „The protective threshold might vary among virus variants [20].“

Reviewer 2 Report

This paper investigated early (<100 days) and late (>=100 days) antibody response to SARS-CoV-2 vaccination in the hemodialysis patients in comparison to the healthy control group. The concept and the strategy of analysis are significant and valuable.

I will ask some questions, recommendations, and revisions to accept for publication.

First of all, the title "Long-term humoral response after SARS-CoV-2 vaccination in hemodialysis patients" seems not appropriate. It may lead to the misunderstanding that long-term humoral response for SARS-CoV-2 is maintained in the hemodialysis patients. As both of early and late humoral response to SARS-CoV-2 vaccination is impaired in the presented data, it is better to change the title. Bucause early humoral response is poorer in the hemodialysis patients, it is difficult to conclude that long-term humoral response is particularly impaired thanshort-term response from the presented data.

As discussed in the text and reported in the literature, anti-SARS-CoV-2 antibody titer after the completion of twice-vaccination peaks at 2-4 weeks, and rapidly decreases after 2-6 months. In this sense, the titer of antibody is profoundly affected when it is examined. It is better to show the median and range of T1 and T2 in the both cohorts.

I think it is an interesting observation that the dialysis patients with COVID-19 infection prior to vaccination showed a more profound long-term vaccine response.  Author should cite the reference to "COVID-19 vaccine, unlike hepatitis B and influenza vaccination, elicit a substantial antibody response in the dialysis population", and discuss this issue in the light of the stronger antibody response to SARS-CoV-2 vaccination in the hemodialysis patients with prior COVID-19 infection.

Author Response

Reviewer 2:

We thank the reviewer for the valuable comments and suggestions.

This paper investigated early (<100 days) and late (>=100 days) antibody response to SARS-CoV-2 vaccination in the hemodialysis patients in comparison to the healthy control group. The concept and the strategy of analysis are significant and valuable.

I will ask some questions, recommendations, and revisions to accept for publication.

First of all, the title "Long-term humoral response after SARS-CoV-2 vaccination in hemodialysis patients" seems not appropriate. It may lead to the misunderstanding that long-term humoral response for SARS-CoV-2 is maintained in the hemodialysis patients. As both of early and late humoral response to SARS-CoV-2 vaccination is impaired in the presented data, it is better to change the title. Because early humoral response is poorer in the hemodialysis patients, it is difficult to conclude that long-term humoral response is particularly impaired than short-term response from the presented data.

  • We changed the title as requested to „Diminished short- and long-term antibody response after SARS-CoV-2 vaccination in hemodialysis patients.“

As discussed in the text and reported in the literature, anti-SARS-CoV-2 antibody titer after the completion of twice-vaccination peaks at 2-4 weeks, and rapidly decreases after 2-6 months. In this sense, the titer of antibody is profoundly affected when it is examined. It is better to show the median and range of T1 and T2 in the both cohorts.

  • We agree with the reviewer that the optimal presentation is difficult to address. All data were presented as median (median (Q1, Q3 in the table). The individual data can be seen in figure 2.

I think it is an interesting observation that the dialysis patients with COVID-19 infection prior to vaccination showed a more profound long-term vaccine response.  Author should cite the reference to "COVID-19 vaccine, unlike hepatitis B and influenza vaccination, elicit a substantial antibody response in the dialysis population", and discuss this issue in the light of the stronger antibody response to SARS-CoV-2 vaccination in the hemodialysis patients with prior COVID-19 infection.

  • We included the reference [Chang et al Vaccines 2021, Kufta et al. Hemodial In 2019, Udomkarnjananun et al. J Nephrol 2020] and discuss this issue in more detail.

“In general, an altered and weaker immune response has been shown for other vaccines like influenza or hepatitis B [28, 29]. Regarding the latter, this led to strategies of identifying individuals at risk by regular measurement of antibody status and of using increased doses or booster vaccinations [30]. A correlation between responsive-ness to hepatitis B vaccination and seroconversion after SARS-CoV-2 vaccination could not be demonstrated so far [14, 16]. It remains unclear if the spike protein is more im-munogenic than the HBs antigen or if mRNA vaccines cause increased immunogenicity. “

Reviewer 3 Report

Authors reported that hemodialysis patients showed a significantly stronger decline of SARS-CoV-2 vaccination antibody titers within 6 months, compared to controls, supports the need for booster vaccinations in order to foster a stronger and persistent antibody response.

Although this manuscript is potentially interesting, several issues arise.

  1. Old aged peoples have lower response in comparison to younger peoples. The age in the patients with hemodialysis was older than that in control. The age of two groups should be adjusted.
  2. The underling diseases in the patients with hemodialysis should be explained.
  3. Control peoples should be explained.
  4. Weight and BMI may be useful.
  5. The stage of COVID-19 infection should be shown.
  6. When were the patients with hemodialysis infected with COVID-19 after vaccination?
  7. The side effects may be helpful.

Author Response

Reviewer 3:

We thank the reviewer for the valuable comments and suggestions.

Authors reported that hemodialysis patients showed a significantly stronger decline of SARS-CoV-2 vaccination antibody titers within 6 months, compared to controls, supports the need for booster vaccinations in order to foster a stronger and persistent antibody response.

Although this manuscript is potentially interesting, several issues arise.

  1. Old aged peoples have lower response in comparison to younger peoples. The age in the patients with hemodialysis was older than that in control. The age of two groups should be adjusted.
  • We agree with the reviewer. We added the following sentence: „Patients in the hemodialysis group were significantly older than controls. In a regression analysis, age could be excluded as a potential confounder.“
  1. The underling diseases in the patients with hemodialysis should be explained.

This is a retrospective study. History of cancer, diabetes and the use of immunosuppression as well as transplantation were recorded and are presented in table 1. Also, the involved dialysis centers present the average of the German dialysis population. Due to the study design we are not able to retrieve more information.

  1. Control peoples should be explained.
  • We added the following information to the Material and Methods (2.1 Study setting): "We did not exclude patients with prior COVID-19 infection or immunosuppressive therapy from analysis. The control group consisted of patients without dialysis treated at the same outpatient centers (patients with chronic kidney disease, patients on LDL apheresis, kidney transplant patients and health care workers during their regular medical visits).
  1. Weight and BMI may be useful.
  • We added the BMI values which were only collected on the hemodialysis group.
  1. The stage of COVID-19 infection should be shown.
  • Due to the study design, we only included patients who survived their COVID 19 infection prior to vaccination. 13 out of 34 hemodialysis patients with a COVID infection prior to vaccination were admitted to hospital, 4 of them to the intensive care unit. We included this information in the paper (Results 3.1. Demographic and clinical data).
  1. When were the patients with hemodialysis infected with COVID-19 after vaccination?

Two patients contracted a COVID infection 2 months after  complete vaccination. In two patients, we were not able to define the precise point of time. Infections occurred prior to the emergence of the Omicron variant.

  1. The side effects may be helpful.
  • This was a retrospective study. Side effects of the vaccination were not recorded and can therefore not be presented.

Round 2

Reviewer 3 Report

This revised manuscript has been sufficiently improved. I have no further comment.